# Interaction Between Early Meals (Big-Breakfast Diet), Clock Gene mRNA Expression, and Gut Microbiome to Regulate Weight Loss and Glucose Metabolism in Obesity and Type 2 Diabetes

**DOI:** 10.3390/ijms252212355

**Published:** 2024-11-18

**Authors:** Daniela Jakubowicz, Yael Matz, Zohar Landau, Rachel Chava Rosenblum, Orit Twito, Julio Wainstein, Shani Tsameret

**Affiliations:** 1Endocrinology and Diabetes Unit, Wolfson Medical Center, Sackler Faculty of Medicine, Tel Aviv University, Holon 58100, Israel; 2Institute of Biochemistry, Food Science and Nutrition, The Robert H. Smith Faculty of Agriculture, Food and Environment, The Hebrew University of Jerusalem, Rehovot 76100, Israel

**Keywords:** circadian clock genes, gut microbiome, big-breakfast diet, weight loss, type 2 diabetes

## Abstract

The circadian clock gene system plays a pivotal role in coordinating the daily rhythms of most metabolic processes. It is synchronized with the light–dark cycle and the eating–fasting schedule. Notably, the interaction between meal timing and circadian clock genes (CGs) allows for optimizing metabolic processes at specific times of the day. Breakfast has a powerful resetting effect on the CG network. A misaligned meal pattern, such as skipping breakfast, can lead to a discordance between meal timing and the endogenous CGs, and is associated with obesity and T2D. Conversely, concentrating most calories and carbohydrates (CH) in the early hours of the day upregulates metabolic CG expression, thus promoting improved weight loss and glycemic control. Recently, it was revealed that microorganisms in the gastrointestinal tract, known as the gut microbiome (GM), and its derived metabolites display daily oscillation, and play a critical role in energy and glucose metabolism. The timing of meal intake coordinates the oscillation of GM and GM-derived metabolites, which in turn influences CG expression, playing a crucial role in the metabolic response to food intake. An imbalance in the gut microbiota (dysbiosis) can also reciprocally disrupt CG rhythms. Evidence suggests that misaligned meal timing may cause such disruptions and can lead to obesity and hyperglycemia. This manuscript focuses on the reciprocal interaction between meal timing, GM oscillation, and circadian CG rhythms. It will also review studies demonstrating how aligning meal timing with the circadian clock can reset and synchronize CG rhythms and GM oscillations. This synchronization can facilitate weight loss and improve glycemic control in obesity and those with T2D.

## 1. Introduction

The endogenous circadian clock coordinates most of the energy and glucose metabolism processes over approximately 24-h cycles. Circadian clock genes (CGs) act as oscillators capable of anticipating daily fluctuations in environmental conditions, such as light-dark cycles and recurring feeding–fasting patterns [1,2,3]. Circadian CGs and meal timing interact with and entrain each other to assign food intake in the most appropriate temporal sequence [4,5,6,7,8,9].

Preclinical and clinical research has demonstrated that consuming most daily calories and carbohydrates (CH) earlier in the day, particularly with a high-energy breakfast, exerts a powerful resetting effect (known as the “zeitgeber effect”) on the CG network. It synchronizes circadian CG expression and regulation of the metabolism throughout the day in rodents [10,11,12,13,14,15,16] and CG mRNA expression in humans [17,18,19,20], as assessed in peripheral white cells [17,18,19] and in adipose tissue [20], underscoring the crucial role of meal timing on CG entrainment and metabolic efficiency

As a result of circadian regulation, calories and especially carbohydrates (CH) consumed in the morning hours are utilized more efficiently, providing several benefits for weight loss and glucose metabolism [6,21,22,23,24,25,26,27,28]. Furthermore, nutrient intake in the early hours of the day is associated with enhanced insulin-driven CG entrainment [29], greater insulin sensitivity and β-cell responsiveness [30,31], and increased GLUT4 activity and insulin-dependent muscular glucose uptake [31,32]. There is greater glucagon-like peptide 1 (GLP-1) post-meal response [33], less hepatic glucose production [34,35], increased fat loss by increasing fat oxidation [36], and enhanced diet-induced thermogenesis (DIT) after a meal consumed in the morning versus in the afternoon or evening hours [28,37,38,39].

Many studies have proved a significant effect of breakfast consumption versus omission on circadian CG expression and metabolic outcomes. Studies in rodents [16,40,41] and humans [17,42,43] showed that the omission of or delayed breakfast is directly associated with disrupted circadian CG expression [16,17] and increased body weight, appetite, higher overall glycemia, and HbA1c [44,45,46,47]. Instead, the consumption of breakfast has been associated with the upregulation of pivotal CG expression, better glucose, insulin, and GLP-1 response after subsequent meals [17,44], and better control of overall glycemia, body weight, and appetite [18,21,26,27]

Some authors, however, reported similar or even worse detrimental effects caused by skipping dinner than breakfast [48]. In addition, the adverse effects of the omission of meals need to be analyzed with caution, as there might be causality confounders, such as socioeconomic status, age, presence of T2D, and other lifestyle factors that may add collateral influence to the health outcomes [49,50] and on glycemic levels and HbA1c [51]. A discussion of the differences between studies on the effects of skipping meals is detailed in Section 5.1.

The first daily meal, breakfast determines circadian CG phasing of peripheral CGs in rodents [16,42]. Some preclinical studies have shown that time-restricted eating (TRE), defined as limiting daily food intake to a 6–10 h period during the active phase, as opposed to ad libitum (AL) consumption throughout the day and night without any time restrictions, can restore robust CG oscillation and improve glucose control and metabolism in mouse models of circadian disruption, aging, and diet-induced obesity (DIO) [10,11,12,13,14,15,16,40,41]. In humans, however, only a few clinical reports have shown the effect of TRE on CG expression. One study on TRE conducted in healthy participants, compared in a 4-day randomized crossover trial the effects of early time-restricted eating (early TRE) with an eating window between 8:00 and 14:00 versus an eating window from 8:00 to 20:00 (control schedule). The early TRE led to upregulation of circadian CG markers and improved 24-h glucose levels assessed with continuous glucose monitoring (CGM) [19]. However, several studies describe a similar metabolic advantage of early and late time-restricted eating [9,23,41,43]. Indeed, in a cross-sectional study conducted on 3813 participants, both early TRE (05.00–15.00) and late TRE (11.00–21.00) showed similar diminished risk of developing non-alcoholic fatty liver disease (NAFLD) [43]. TRE in animals and humans and its effects on CG expression and GM composition, including clinical trials conducted during Ramadan, are discussed in more detail in Section 5.3.

Recently, growing evidence has revealed that the community of microorganisms in the gastrointestinal tract, known as the gut microbiome (GM), also undergoes daily oscillation in relative abundance and function in mice [11,12,13,14,15,52,53,54,55,56] and humans [52,57,58,59,60,61].

These GM microorganisms entrain upon non-light-related cues, primarily the timing of food intake, which significantly coordinates GM oscillation and the release of GM-derived active metabolites. These metabolites further serve as non-canonical drivers of circadian CGs, influencing the metabolic response to food intake [53,54,62,63,64,65,66,67,68,69,70,71,72,73].

Irregular eating patterns, such as skipping breakfast, overeating at night, or eating all day and night (ad libitum) without any specific feeding–fasting schedule, can disturb circadian CGs and the rhythm of the GM (dysbiosis) in a reciprocal manner, contributing to metabolic dysfunction, obesity, and T2D, as reported in animal models [10,11,12,13,14,15,16,40] and clinical studies [17,18]. In contrast, shifting meal timing to the early part of the day or following an eating schedule restricted to a 6–10 h period during the active phase can upregulate CG expression and positively impact GM diversity and composition in rodents [11,12,13,14,16] and humans [17,18]. Therefore, the synchronized effect of meal timing on the circadian CGs and GM can play a pivotal role in influencing energy balance and metabolism [17,18,19,20,21,22,23,24,25,62,63,64,65].

This review examines the interaction between circadian CG rhythms, GM oscillations, and meal timing. It will cover studies demonstrating how aligning meal timing with the circadian clock can reset and synchronize circadian CGs and GM oscillations. We will also explore the potential implications of aligned dietary timing to reset circadian CGs and the GM as potential interventions for regulating metabolism in patients with obesity and T2D.

## 2. Circadian Clock System

### 2.1. Circadian Clock Genes: Regulation of Energy and Glucose Metabolism

The endogenous clock is a network of molecular clocks with a central or master clock located within the hypothalamus’s suprachiasmatic nucleus (SCN) [1,4]. The central clock is synchronized with the day–night cycle and transmits information regarding light–dark transitions to cells across the body via neuronal signals, temperature, behavioral modifications, the wake–sleep cycle, and hormonal signaling, primarily adrenal glucocorticoids [1,2,3,4]. The central clock generates an internal rhythm of approximately 24 h through the action of self-sustained clock genes found in most peripheral tissues, such as β-cells, intestinal L-cells, adipose tissue, and gut microbiota. However, the primary time giver or “zeitgeber” for peripheral clocks is not light, but the timing of meals and the feeding–fasting schedule [2,3,4,5,74].

The circadian CGs in peripheral tissues coordinate numerous biological processes, including metabolism, behavior, immunity, and the composition and function of the gut microbiota. Their primary role is to synchronize these processes with the diurnal fluctuations in environmental conditions [1,4,53,54]. Numerous enzymes and hormonal functions implicated in energy and glucose metabolism exhibit circadian oscillation and are regulated by circadian CGs. These encompass β-cell secretion and insulin sensitivity [30,31], insulin-dependent muscular glucose uptake [31,32], hepatic glucose production [34,35], and the postprandial response of glucagon-like peptide 1 (GLP-1) [33]. The circadian CGs also regulate the oscillation and composition of GM and the production of active metabolites derived from gut microbiota [12,14,52,53,54].

### 2.2. Synchronization of Central and Peripheral Clock Genes

Achieving metabolic homeostasis requires synchronization of the master clock controlled by light and peripheral CGs, which are synchronized with the time of food intake [5,6,7,8,9,23,31,41]. The timing of food intake coordinates the peripheral CGs through various feeding-related hormones and signals, including absorbed nutrients [75,76,77,78]. This synchronization allows peripheral clocks to predict the release of metabolic hormones and enzymes right before food intake at a specific, predetermined time of day. This results in the timed regulation of digestive, absorptive, and metabolic functions [22,31,33,34] (Figure 1).

The first meal of the day, breakfast, has a powerful resetting effect (zeitgeber effect) on the body’s clock gene network. Therefore, consuming most calories early in the day provides a metabolic advantage in animal models [10,16,40,76] and humans [17,18,23,76]. The molecular clock drives the rhythmic fluctuations of metabolite levels in each tissue type. These metabolic products and hormones serve as essential feedback inputs into the circadian CG network, creating a bidirectional feedback loop and reciprocal control between cellular metabolic activity and circadian CGs [1,3]. Noteworthy is the fact that the interaction between circadian CGs and the diurnal oscillations in the GM mutually regulate each other [14,52,63,64,65] Figure 1.

Moreover, GM-derived active metabolic products serve as an essential input into the circadian CG network, thereby creating a bidirectional feedback loop control between GM metabolic activity and circadian CGs [14,52,63] Figure 1.

### 2.3. Circadian Clock Molecular Mechanism

The molecular clock mechanism involves cell-autonomous transcription–translation feedback loops. These loops are based on the degradation of regulatory enzymes and different transcription factors, allowing for recurring cycles. The primary input comes from light, which synchronizes the central clock and many feedback loops in peripheral tissues. This synchronization in peripheral CGs is controlled mainly by meal timing. It enables repeated cycles to maintain oscillations in gene expression [2,4].

At dawn, at the beginning of the cycle, two transcription activators, *CLOCK* (circadian locomotor output cycles kaput) and *BMAL1* (brain and muscle *ARNT-like 1*), act as positive elements in the feedback loop. *CLOCK* and *BMAL1* dimerize to form a CLOCK–BMAL1 complex, which binds to the E-box promoter element to activate the transcription of period genes *(PER1*, *PER2*, and *PER3*) and cryptochrome genes (*CRY1* and CRY2). After translation, the resulting PER and CRY proteins dimerize. They accumulate in the cytoplasm during dusk and then, during the night, are translocated back into the nucleus to interact with the CLOCK–BMAL1 complex. This interaction directly suppresses their own transcription, thus generating a new cycle that recurs every 24 h [2,3,4,77,78].

The blockage of CLOCK–BMAL1 is reversed before dawn by casein kinase I epsilon (CKIε), thereafter a new 24-h transcription cycle re-starts, just before waking up. CKIε also phosphorylates and partly reactivates *BMAL1-*driven transcription [2,4,77].

In a separate regulatory process, the CLOCK–BMAL1 complex controls the transcription of the retinoic acid receptor-related orphan receptor (*RORα*) and the nuclear receptor *REV-ERBα*. The resulting proteins serve as positive and negative regulators of *BMAL1* transcription, respectively, thereby contributing further to the clock’s 24 h oscillation [79] (Figure 2).

The CLOCK–BMAL1 heterodimer is also responsible for mediating the transcription of tissue-specific output CGs. This includes *PER*s, *CRY*s, *REV-ERB*s, and *ROR*s, along with *PPARγ* coactivator 1α (*PGC-1α*), *SIRT1*, and other transcriptional elements. These elements promote downstream expression of several tissue-specific proteins, hormones, and enzymes, relaying the clock information to cellular processes like β-cell secretion, muscular glucose transporter type 4 *(*GLUT4) activity, hepatic glycogenolysis, and gluconeogenesis [34,35,78,79,80,81,82,83]. During the active phase (fed state), the expression of *BMAL-1*, *RORα*, and *SIRT1* plays a crucial role in regulating the circadian changes of insulin sensitivity [81,83], β-cell responsiveness [34], postprandial GLP-1 secretion [33], muscular GLUT-4 activity, and glucose uptake [31]. During the nocturnal resting phase, the expression of *REVERBα*, *RORα*, and *SIRT1* in the liver promotes glycogenolysis enzyme glucose 6-phosphatase (HG6-P) in the first part of the nocturnal resting phase and phosphoenolpyruvate carboxykinase (PEPCK) in the gluconeogenesis pathway during the second part of the nocturnal phase [31,32,35,79]. In addition, *BMAL1* transcription of *PPARα* and *PGC-1α* in adipose tissue plays a role in lipid metabolism (lipogeneses and nocturnal lipolysis) [46,47,48,79,80] (Figure 2).

Nutrient-signaling molecules are potent regulators of clock genes in peripheral tissues. Feeding results in the activation of the insulin-phosphorylated protein kinase B (pAKT)–mammalian target of rapamycin (mTOR) pathway, which increases the stability and translation of *PER* [20,29,84], whereas fasting activates AMP-activated protein kinase (AMPK) and nicotinamide phosphoribosyltransferase (NAMPT), reducing the stability and transcription of CRY and *PER* [20,84,85]. Furthermore, adenosine monophosphate-activated protein kinase (AMPK) positively interacts with SIRT1, improving β-cell viability, GLUT4 expression, and translocation, enhancing muscular glucose uptake [81,82,83,84,85].

Therefore, meal timing is an entraining cue for peripheral clocks, and mistimed eating with shortened overnight fasting periods dampens peripheral CGs in mice [11,12,14] and in humans [17,18,19,86], leading to deleterious effects on lipid and glucose metabolism.

## 3. Gut Microbiome

### 3.1. Gut Microbiome Composition and Its Link with Obesity and T2D

The gut microbiome (GM) consists of trillions of microorganisms residing in the gastrointestinal tract. It plays a crucial role in numerous metabolic processes, including food digestion, secretion of immunostimulatory molecules, chemical reactions in the intestine, and production of metabolites [52,53,54,55,56,57,58,59,60,61,62,63,64,65,66,67,87,88,89,90].

Over 60% of the intestinal bacteria’s abundance, composition, and function display circadian oscillations coordinated by circadian CG rhythms and the feeding–fasting schedule [52,64,68,69,87]. The GM is a key player in the interaction between meal schedules and circadian CGs [14,52,62,63]. The metabolic signals released by the GM, in synchrony with the feeding–fasting cycle, can modulate circadian CG expression, playing a crucial role in metabolic homeostasis [13,15,64,70,71].

GM primarily includes four main families (phyla): *Firmicute*s, *Bacteroidete*s, *Proteobacteria*, *and Actinobacteria*. The Firmicutes account for 64%, followed by *Bacteroidete*s at 23%, *Proteobacteria* at 8%, and *Actinobacteria* at 3% [88,90,91,92,93]. Greater GM diversity is generally associated with improved lipid profiles, anti-inflammatory cytokines, liver enzymes, and other indicators of better energy and glucose metabolism [53,60,61,62,63,87,88]. Evidence also suggests that disruption of stable GM communities may increase the prevalence of proinflammatory conditions, including obesity and T2D [53,68,92,93,94].

A decreased ratio of *Bacteroidete*s to *Firmicute*s has been observed in obese (ob/ob) mice, with a 50% reduction in the abundance of Bacteroidetes and a proportional increase in *Firmicute*s compared to their lean siblings [14,92,95].

Humans with obesity or T2D also have less rhythmic gut microbiota and lower microbial diversity than controls [52,93,96,97,98]. However, the specific types of bacteria that lose rhythmicity in obesity are different from those in T2D, suggesting that obesity and T2D have different effects on microbiome rhythmicity [52,57,58]. In obese humans, microbial taxon depletion and dispersed microbiota abundance consist of a reduced *Bacteroidete*s-to-*Firmicute*s phylum ratio [93,94,96,97]. Moreover, low bacterial counts of *Bacteroidete*s in obese individuals are associated with atherosclerosis, dyslipidemia, higher insulin resistance, and diabetes when compared to those with high bacterial counts [97,99]. T2D is also associated with variations in intestinal bacterial abundance and a significant decline in GM diversity [57,58,59,91,97,98,99,100,101]. This consists in an increase in *Bacteroidete*s species versus a decrease in the *Firmicute*s phylum compared to controls. Furthermore, the *Bacteroidete*s-to-*Firmicute*s ratio is associated with increased plasma glucose following an oral glucose load [59,90,100,101]. Collectively, these data clearly suggest that the composition and function of the GM may contribute to glycemic regulation and insulin sensitivity [58,94,97,100,101].

### 3.2. Synchronizing the Eating–Fasting Schedule with the Rhythms of the Gut Microbiome

The GM displays circadian oscillations synchronized with circadian CGs and aligned with the eating–fasting schedule [53,54,62,63,64,65,66,67,68].

About 15% of bacterial species and 24% of GM operational taxonomic units (OTUs) exhibit marked diurnal oscillation in composition, abundance, diversity, and epithelial adherence within a 24 h cycle [63,66,67,68]. The GM’s abundance, diversity, and function are higher during the active or feeding phase and lower during the resting or fasting phase [63,66,67,102,103,104,105,106]. In the active phase, there is an increase in GM taxonomic and functional composition, alpha diversity, and bacterial attachment to the intestinal epithelium. Bacterial motility, mucus degradation, and gut microbiota antimicrobial activity also increase during the active phase [63,64,102,103,104,105].

Hence, it is evident that GM oscillations are influenced by the feeding–fasting pattern [64]. The dominant phyla of the gut microbiota, *Firmicute*s and *Bacteroidete*s, exhibit diurnal rhythmicity in both mice and humans. *Firmicute*s are most abundant during feeding, reaching their peak near the end of the dark–feeding active phase and decreasing during the light–fasting inactive phase. On the other hand, *Bacteroidete*s peak during fasting and decrease during feeding [12,14,57,65,103].

Research has demonstrated that dietary intervention (DI) can significantly impact the oscillation of the GM. Mice on a normal or non-high-fat diet exhibit cyclical fluctuations in most GM species. However, mice given an ad libitum high-fat diet (AL-HFD), a model of diet-induced obesity, show a loss of diurnal feeding-related oscillation in their GM [12,13,14,41]. The HFD decreases the expression of the regenerating islet-derived protein (Reg3γ) in the intestinal epithelial cells (IECs), disrupting the rhythms of gut microbiota and promoting metabolic dysfunction [14,65,89]. A key finding in these studies is that feeding patterns influence the composition and oscillation of the GM even when animals are fed the same diet but with different meal schedules [11,12,14].

The coordinated diurnal CG and GM rhythmicity provides evidence of the microbiome-dependent mechanism for common metabolic disturbances in humans with aberrant feeding rhythms, as seen in shift workers and frequent flyers [63,64].

### 3.3. Mechanisms of the Gut Microbiome’s Role in Energy and Glucose Metabolism

The community of gut microbiota can influence energy and glucose metabolism through various mechanisms, including (1) production of short-chain fatty acids (SCFAs), (2) regulation of bile acid metabolism, (3) control of inflammation, cytokine release, and production of bacterial lipopolysaccharides (LPSs), (4) maintenance of the integrity of the intestinal epithelial barrier, (5) regulation of bacterial adhesion to the intestinal mucosa, and (6) modulation of metabolic clock gene expression (Figure 2).

#### 3.3.1. Production of Short-Chain Fatty Acids (SCFAs)

Short-chain fatty acids (SCFAs) are organic acids containing two to six carbon atoms and are the main products of the gut microbiota’s anaerobic fermentation of indigestible dietary fibers. Butyrate, acetate, and propionate make up 95% of the SCFAs produced by the gut microbiota [105,106,107,108,109,110,111,112,113,114]. The *Firmicute*s phylum is the major butyrate producer, while the *Bacteroidete*s phylum primarily produces acetate and propionate [87,88,90,91]. SCFAs, especially butyrate, are crucial to metabolic functions [107,108,109].

Butyrate is linked to enhanced gut wall integrity, reduced gut permeability [107], and improved insulin sensitivity and energy expenditure in mice [110]. It also has anti-inflammatory effects, decreasing the release of cytokines and lipopolysaccharides (LPSs), which may in turn alleviate LPS-induced insulin resistance [59,108,111]. SCFAs substantially impact blood glucose metabolism by increasing the expression of GLUT4 through AMP-activated protein kinase (AMPK) activity, enhancing muscular glucose uptake [110,111,112]. High butyrate activity enhances insulin response during an oral glucose tolerance test, indicating better β-cell function [110,112,113]. Butyrate can also reduce glycolysis in skeletal muscle, resulting in secondary accumulation of glucose 6-phosphate and greater muscular glycogen synthesis [110,111].

In summary, butyrate improves insulin sensitivity and β-cell secretion, decreases hepatic glycolysis and gluconeogenesis, increases muscular glycogen synthesis, and lowers plasma fatty acid concentration [58,59,112,113,114]. Notably, butyrate, through a cAMP-dependent mechanism, induces the expression of genes involved in intestinal gluconeogenesis [115] (Figure 3).

Increased delivery of propionate to the intestines has also been linked to improved β-cell function. Furthermore, propionate directly protects human islets by inhibiting cytokine-induced apoptosis [54,116]. Both butyrate and propionate activate cell-surface G protein-coupled receptors (GPRs) and can stimulate intestinal gluconeogenesis, a mucosal process with anti-diabetic and anti-obesity effects [117,118,119]. In fact, administering oral butyrate in db/db mice reduces plasma levels of HbA1c, inflammatory cytokines, and lipopolysaccharides (LPSs), which may improve insulin sensitivity [120].

#### 3.3.2. Regulation of Bile Acid Metabolism

Bile acids (BAs) are potent digestive surfactants that work as emulsifiers, assisting in the digestion and absorption of lipids [54,71]. The GM plays a crucial role in synthesizing, composing, and signaling BAs. In turn, BAs can influence and regulate the gut microbiota. Therefore, there is a mutual link between the GM and BAs. A disrupted GM is linked to unfavorable changes in BA composition and can lead to metabolic diseases such as type 2 diabetes (T2D), obesity, and non-alcoholic fatty liver disease (NAFLD) [121,122,123]. BAs are secreted by the liver as conjugated compounds (primary bile acids) and deconjugated by GM-derived enzymes to form secondary bile acids [54,124,125,126,127,128]. Notably, these GM-derived enzymes are significantly reduced in individuals with T2D [128]. Deoxycholic acid (DCA) and lithocholic acid (LCA) represent the most common secondary BAs, which activate various beneficial glucose metabolism pathways [125,126,127,128].

Secondary BAs interact with farnesoid X receptor (FXR) and G protein-coupled receptor 5 (TGR-5), forming the BA–TGR-5 complex. This complex reduces gluconeogenesis, promotes hepatic glycogen production [100,125,127], and stimulates GLP-1 secretion, increasing glucose-stimulated insulin release [125,126,127]. The BA–TGR-5 complex also stimulates the secretion of the entero-hormone fibroblast growth factor (FGF19) from the ileum [129], which stimulates postprandial hepatic glycogen production, thereby reducing blood glucose concentrations [129]. FGF19 also plays a role in regulating muscle mass and reducing adipose tissue [130] (Figure 4).

#### 3.3.3. Regulation of Inflammation, Cytokine Release, and Lipopolysaccharide Production

Obesity and T2D are characterized by chronic low-grade inflammation and abnormal expression and production of many inflammatory mediators [131,132]. The GM is a crucial regulator of inflammation by controlling endotoxins and lipopolysaccharides (LPSs) [133].

In healthy microbiota symbiosis, the GM effectively limits the excessive release and absorption of inflammatory mediators. On the other hand, an unhealthy GM (dysbiosis) can lead to an excessive release of endotoxins and LPSs, resulting in increased inflammation throughout the body. This is associated with insulin resistance, weight gain, hyperglycemia, and the development of T2D and NAFLD [53,133,134]. Furthermore, reduced production of butyrate by GM in individuals with T2D is associated with low-grade inflammation in the gut [70,100]. Innate Toll-like receptors (TLRs), particularly TLR-4, recognize bacterial fragments and LPSs, triggering the activation of the intracellular signaling pathway, nuclear factor kappa-light-chain-enhancer of activated B cells (NF-κB), and the release of proinflammatory cytokines, which are linked to deficient β-cell secretion and insulin resistance [133,135,136].

GM dysbiosis may increase the release of TNF-α and other proinflammatory cytokines, disrupting glucose metabolism and insulin signaling. Indeed, patients with T2D have elevated levels of TNF-α, which are linked to insulin resistance and islet dysfunction [53,137,138,139,140]. Interleukin 1 (IL-1) is an inflammatory cytokine linked to reduced movement of GLUT-4 to the cell membrane by reducing insulin–IRS-1 signaling, which decreases insulin-stimulated glucose uptake [141]. It was shown that treating T2D patients with an IL-1 receptor antagonist (IL-1RA) and an IL-1β-specific antibody can enhance glucose metabolism and insulin secretion [142]. IL-6, which is also linked to dysbiosis, has been identified as an independent predictor of T2D. IL-6 reduces insulin–IRS-1 signaling, GLUT4 activity, and insulin-stimulated glucose transport [143,144]. Notably, serum levels of LPSs, TNFα, and IL-6 are increased in individuals with T2D [142,143,144] Figure 2.

#### 3.3.4. Preservation of the Integrity of the Gut Epithelial Barrier

One of the GM’s essential roles is preserving gut barrier integrity. It prevents the passage of undesirable pathogens and harmful substances across the mucosal surface [54,145]. In metabolic diseases such as T2D and obesity, gut barrier function is damaged. This can lead to local inflammation and disrupt the interaction between the microbiota and the gut, negatively affecting energy and glucose metabolism [54,145].

GM-derived SCFAs, especially butyrate, have been linked to enhanced gut integrity [146]. Treatment with butyrate increases gut integrity and intercellular adhesion molecules while reducing the release of LPSs. Therefore, butyrate may prevent gut leak and diabetic endotoxemia in db/db mice and may reduce LPS-induced insulin resistance [71,146]. Moreover, as we mentioned before, oral butyrate administration significantly lowered HbA1c, inflammatory cytokines, and LPS plasma levels in db/db mice [128] (Figure 3).

#### 3.3.5. Regulation of Mucosal Bacterial Adhesion to Intestinal Epithelial Cells (IECs)

Diurnal oscillations in the GM are important diet-dependent drivers of circadian CG rhythms and metabolism, ensuring optimal energy balance [53,62,64,89]. GM rhythmicity influences the diurnal variation of mucosal bacteria adhesion to intestinal epithelial cells (IECs). In addition, GM diurnal fluctuation synchronizes circadian CG transcriptome oscillations [53,62,64,89].

The interplay between diet and GM oscillation is complex. It was reported that an antimicrobial peptide (AMP) associated with GM, known as regenerating islet-derived protein 3 gamma (Reg3γ), orchestrates the interaction between diet and GM bidirectionally. Reg3γ is abundantly expressed and secreted in the small intestine and interacts with key ileal microbes to regulate the diet and GM interplay [56,89].

Reg3γ also participates in host defense via its bactericidal activity, preventing mucosal bacteria from adhering to IECs. Reg3γ binds to the peptidoglycan layer on the surface of Gram-positive pathogens, reducing bacterial adhesion [56,89]. Mice that lack Reg3γ exhibit disrupted GM rhythms and increased diurnal adherence patterns of mucosal bacteria to the intestinal epithelium [64,89].

It was also reported that a high-fat diet (HFD) can lead to disrupted GM oscillation and metabolic imbalances. This disruption can cause irregular expression of Reg3γ, affecting the abundance and oscillation of key gut microbes [89,132]. In addition, the gut peptide Reg3γ links the small-intestine GM to regulating energy balance, glucose levels, and gut function. Reg3γ is known to have protective effects against oxidative stress, suppressive the effects of epithelial inflammation and bacterial colonization, and potentially increases insulin sensitivity through its impact on skeletal muscle, thereby enhancing glucose homeostasis [62,64,89] (Figure 2).

#### 3.3.6. Regulation of Clock Gene Expression by Gut Microbiome

Recent studies indicate that the GM influences digestion, absorption, and energy balance and is intricately linked and synchronized with circadian CGs in a bidirectional manner [15,62,63,64]. The GM’s transcriptional activity is highly dynamic, featuring cyclical gene expression and daily fluctuations in its composition and function [147,148,149,150,151,152,153,154,155].

Meal timing, circadian CGs, the GM, the immune system, and metabolism are all interconnected [53,132,133]. It has been recognized that the GM acts as a transducer of dietary cues to regulate circadian CG rhythms and metabolism [147,148]. Therefore, any disruption of the GM, either via environmental manipulations [62,63], a high-fat diet [14,65], or genetic mutation [64,102], may result in loss or disturbed CG oscillations. GM oscillations in the cecum, colon, and stool are driven by the time of feeding and nutrient delivery to the gut over 24 h. However, it was also previously revealed that gut microbes from mice fed via continuous parenteral nutrition still exhibited diurnal oscillations relative to their enterally fed counterparts [65]. This suggests the existence of additional gut signals, whether from CGs or microbially derived signals, that can play a role in GM rhythmicity [65].

The synchronization between the GM and circadian CGs is mediated by hundreds of GM-derived metabolites that oscillate in coordination with the meal schedule [15,63,148,151,152,153]. The GM entrains upon non-photic cues, primarily the timing of food intake, which significantly coordinates GM oscillation and the release of GM-derived active metabolites, which further interact to regulate the peripheral circadian CGs and metabolism [106,147,149,150,152]. Microbial components such as lipopolysaccharides (LPSs) and metabolites derived from the GM, such as SCFAs and unconjugated BAs, may interfere with the expression of molecular CGs, thereby affecting a variety of circadian processes and whole-body metabolism [147,149,153,154,155].

The signals derived from the gut microbiota, particularly GM-derived SCFAs, exhibit daily fluctuations and significantly influence the circadian pattern of CG expression [65,106,156]. A recent report revealed that serum from mice orally injected with butyrate synchronized cultured hepatocyte PER2/BMAL1 expression rhythm. The intraperitoneal injection of butyrate did not influence mouse liver PER2/BMAL1 expression ratio. This suggests that GM-derived SCFAs have a pivotal influence on peripheral CG entrainment [65]. In another study, it was discovered that in mice, three days’ oral administration of a combination of SCFAs (acetate, butyrate, and propionate) led to changes in *PER2–LUC* gene oscillation in liver and kidney tissues [106].

GM-derived SCFAs, especially butyrate and acetate, are associated with upregulating circadian CGs, such as *PER2*, *PER3*, and BMAL1. This upregulation further amplifies the effects of microbial-derived SCFAs and their beneficial impact on metabolism [65].

On the other hand, when mice were fed a high-fat diet (HFD) and experienced a decrease in GM-derived SCFAs, this caused disruptions in circadian CG oscillations compared to mice on a low-fat or normal-fat diet (NFD) [12,14,65]. These findings suggest that diet affects the production and fluctuations of GM-derived SCFAs, subsequently influencing circadian CG expression [14,65,147] (Figure 3).

The timing of meals and the feeding–fasting schedule coordinate the CG-controlled oscillatory production of GM-derived SCFAs. These GM-derived SCFAs are released into the bloodstream in a circadian manner [15,64,117,118]. Once in the bloodstream, the GM-derived SCFAs can influence CG-controlled rhythmic glucose metabolism by binding and activation of specific cell-surface G protein-coupled receptors (GPR43/41) in various tissues, including the liver, white adipose tissue, skeletal muscle, pancreas, intestine, and immune cells [105,117]. These cell-surface G protein-coupled receptors, GPR43/41, serve as signaling molecules in the metabolic communication between the GM and peripheral tissue CGs and metabolism [117,118]. The timing of diet meals influences the concentration of GM-derived SCFAs in feces and the expression of GPR43/41 in the colonic muscle layer. As a result, GM-derived SCFAs may regulate colonic circadian motility in mice [105].

Activating cell-surface GPR43/41 also stimulates the secretion of the incretin GLP-1 from enteroendocrine L-cells, which enhances glucose-induced insulin secretion, protects β-cells from apoptosis, and promotes β-cell proliferation [118,119]. Moreover, cell-surface GPR43/41 activation by GM-derived SCFAs also links GM metabolic activity with energy metabolism. In fact, it was shown that mice lacking GPR43/41 became obese on a regular diet, whereas mice with overexpression of GPR43/41 in their adipose tissue remained lean even when fed a high-fat diet [71,118].

Unconjugated bile acid (BA) content is another factor by which the GM may influence the expression of peripheral CG rhythms [123,148,149]. Microbes in the gut lumen deconjugate bile acids (BAs) into unconjugated BAs, which influence the amplitude and periodicity of circadian CG oscillation in the ileum, colon, and liver [123,148,149]. In a study using a model of intestinal epithelial cell lines with synchronized circadian rhythms, it was reported that treatment with unconjugated BAs significantly altered the amplitude and cyclic behavior of circadian CGs, including *CLOCK*, *BMAL1*, *PER*, *CRY*, *RORA*, and *REV-ERBα*, in the ileum, colon, and liver. This treatment also led to significant changes in the expression of hepatic regulators of circadian rhythm and associated genes (*PER*2, *PER*3, and *CRY*2). The data demonstrate a potential mechanism for microbe–host cross talk that significantly impacts host circadian CG expression [149,152].

Lipopolysaccharides (LPSs) are also involved in communication between gut microbes and circadian CGs, synchronizing the metabolic response to diet. This suggests another mechanism for cross-talk interaction between circadian CGs and the GM [54,149,150,151,152] (Figure 4).

In summary, preclinical and clinical research have shown that disrupted CGs and GM composition are linked to a higher risk of obesity, metabolic syndrome, hyperglycemia, and hyperlipidemia [14,102,150,151,152,153,154,155,157,158]. Disrupted CGs and GM were also evident in Goto–Kakizaki rats, a model for nonobese type 2 diabetes [159], and in Clock(Δ19)-mutant mice [160]. Moreover, circadian CG mutations may lead to GM dysbiosis and promote further metabolic disturbances [159]. Similarly, in humans, disrupted GM composition and CG expression are associated with obesity, hyperglycemia, and T2D [17,18,101,150,157].

In the next section, we will discuss how misaligned meal timing or disruptions in daily routines, such as switching meal timing from a light–dark cycle, can disrupt CGs and may lead to an unbalanced GM. This in turn can cause the asynchrony of pivotal circadian CGs, leading to metabolic disturbances.

## 4. Meal Timing Alignment and Misalignment: Effect on Clock Genes and Gut Microbiome

As mentioned above, the meal schedule exerts a pivotal synchronizing influence on CG rhythms [5,6,9,14,15,16,17,18] and GM composition [11,12,14,62,63,64,65,101,102,103]. As there is reciprocal communication between circadian CGs and the GM, the focal point of the cross-talk network between CGs and the GM is the meal schedule, which determines alignment or misalignment. Indeed, diet is one of the environmental factors that most influences both circadian CGs and GM composition and function [53,54,62,63,64,65,66,67,68,69,70,71].

If meal timing is not synchronized with the light–dark cycle, “meal timing out of phase” may lead to disruption of circadian CG expression, the GM, and metabolism [11,12,14,62,63,64,65,101,102].

Circadian CG disruption has been reported in rodents with irregular meal schedules, such as omitting the first meal of the day [16,40] or eating or snacking all day and night without any time restriction or “eating window”, and is associated with altered metabolism, weight gain, and hyperglycemia [10,11,12,13,14,41,63,64,65]. Other studies in rodents have shown that similar irregular misaligned meal timing—such as skipping breakfast or eating continuously throughout the day—can also disrupt GM balance, causing GM dysbiosis, which can further induce asynchrony of circadian CG expression and metabolic disturbances [14,62,63,64,65,102,154,155,160].

Moreover, an arrhythmic GM caused by misaligned meal timing in mice may alter the symbiosis between the GM and intestinal epithelial cells (IECs), which can lead to asynchrony of CGs *RORα* and *RevErbα* and altered circadian genes within IECs, which have been associated with a prediabetic syndrome due to ileal corticosterone overproduction consequent to clock disruption [151,152]. It was also reported in mice that meal timing-induced GM oscillation regulates the diurnal rhythms in host metabolism through histone deacetylase 3 (HDAC3), which integrates GM and circadian CG cues to regulate diurnal metabolic rhythms [153].

Disruption of the GM and circadian CGs caused by misaligned meal timing can also negatively affect metabolism by altering the release of SCFAs, as reported in mice [113,114,115,116,117,141] and in obese and T2D individuals [116]. Moreover, altered CGs and GM composition driven by misaligned meal timing are linked to insulin resistance and T2D through the activation of inflammatory pathways in mice [133,134,135,141] and in humans to metabolic syndrome and T2D [131,132,142,143,144]. Notably, reduced GM-driven SCFAs induced by misaligned meal timing have been associated with increased gut permeability, higher levels of plasma LPSs, and elevated proinflammatory cytokines such as IL-6, IL-1, TNFα, and hs-CRP, as shown in preclinical [53,131,137,138,139,140,141] and clinical studies [131,132,140,144]. These changes link misaligned meal timing with insulin resistance, obesity, and T2D [66,135,136,137,140,154].

In contrast, aligned meal timing can restore CG and GM oscillation and increase the GM-driven release of SCFAs while reducing several proinflammatory factors, thereby improving energy and glucose metabolism [14,66,71,147].

It has been documented that aligned meal timing and a balanced GM suppresses insulin-mediated fat accumulation via binding GM-derived SCFAs with GPR43/41 cell-surface receptors [117,118]. The activation of GPR43/41 also enhances GLP-1 secretion, reducing fat accumulation [119]. In contrast, the decreased release of GM-derived SCFAs triggered by misaligned meal timing and GM dysbiosis may enhance insulin-mediated fat accumulation [117,118] and reduce GLP-1 secretion, further favoring fat accumulation, by reducing binding of GM-derived SCFAs with GPR43/41 [119].

Misaligned meal timing and the resulting GM dysbiosis can also affect the fluctuation of primary and secondary bile acids (BAs). This disruption leads to deficient secretion of FGF19, which contributes to insulin resistance and hyperglycemia in mice and humans [87,128,129]. However, aligning meal timing can help restore GM oscillation and the cyclical changes in primary and secondary BAs and facilitate the interaction between FXR and TGR5. This interaction promotes the secretion of FGF19 from the ileum, thereby enhancing energy expenditure, insulin sensitivity, and glucose homeostasis [87,129] (Figure 3 and Figure 4).

Epidemiological studies have shown that irregular meal schedules, which are relevant to individuals working irregular shifts or those who habitually skip breakfast, engage in continuous snacking, or overeat in the evening, are associated with an increase in body weight, hyperglycemia, and high HbA1c [45,46].

In several randomized controlled clinical studies in humans, it was also shown that nonaligned meal timing like continuous snacking all day, the omission of breakfast, or overeating in the evening was associated with obesity, difficulties in losing weight, hyperglycemia, high triglycerides, more appetite, and higher HbA1c [21,24,25,26,27,47,86,161,162]. In contrast, aligned meal timing, i.e., limiting the number of daily meals to two or three with a high-energy breakfast and reduced dinner led to more significant weight loss, better glycemic control, reduced HbA1c, and lower lipid levels and appetite [6,9,17,18,21,25,26,27,161,162]. In line with preclinical studies, increasing evidence shows an association between obesity, metabolic syndrome, and T2D with disrupted CG expression and abnormal GM composition [17,18,101,128,129,132,150,154,157].

Only a few human studies have proved that misaligned meal timing may disrupt CGs and GM in parallel and negatively affect metabolism, e.g., by altering the release of SCFAs, as reported in obese and T2D individuals [116]. Reduced GM-driven SCFAs induced by misaligned meal timing may lead to increased gut permeability, higher levels of plasma LPSs, and elevated proinflammatory cytokines such as IL-6, IL-1, TNFα, and hs-CRP, in several clinical studies [131,132,140,144]. These changes are linked to insulin resistance, obesity, and T2D [66,154]. Indeed, altered CG and GM composition driven by misaligned meal timing through the activation of inflammatory pathways may induce insulin resistance and T2D in humans [131,132,142,143,144].

As the GM suppresses insulin-mediated fat accumulation via binding of the SCFAs with GPR43/41 cell-surface receptors, it was shown that a decreased release of SCFAs triggered by misaligned meal timing and GM dysbiosis may reduce the binding of SCFAs with GPR43/41 and may enhance insulin-mediated fat accumulation, both in rodents and humans [117]. GM dysbiosis caused by misaligned meal timing can also affect the fluctuation of primary and secondary bile acids (BAs), leading to deficient secretion of FGF19 and subsequent insulin resistance and hyperglycemia in mice and humans [87,128,129]. Notably, aligned meal timing can restore GM oscillation, cyclical changes in BAs, and the interaction of FXR with TGR5, which promotes the secretion of FGF19, improving insulin sensitivity and glucose homeostasis in mice and humans [87,129] (Figure 3 and Figure 4).

Clinical studies have also shown that misaligned meal timing, such as skipping breakfast, snacking all day, overeating in the evening, or frequent meals evenly distributed throughout the day, leads to asynchrony in circadian CG expression, and may induce metabolic dysfunction, insulin resistance, weight gain, hyperglycemia, and elevated HbA1c levels [17,18,157]. Clinical trials have reported a circadian CG disruption in healthy and T2D individuals placed on misaligned meal schedules like skipping breakfast [17]. However, breakfast consumption restored CG expression in both groups [17]. Another study in T2D individuals treated with insulin showed that frequent meals evenly distributed throughout the day led to disrupted CG expression. In contrast, high-energy breakfast in the context of only three daily meals upregulated CG expression and led to significant improvement in overall glycemia and HbA1c [18].

Next, we will discuss several preclinical and clinical studies demonstrating how aligned or misaligned meal schedules can affect energy and glucose metabolism by influencing circadian CG, GM, or both.

## 5. Understanding the Impact of Early Meals and Time-Restricted Eating on Circadian CG Expression, Gut Microbiome (GM), and Regulation of Energy and Glucose Metabolism

The GM and molecular circadian CGs play a coordinated role in neural processing, metabolism, adipogenesis, and inflammation. GM and CG interactions are influenced by dietary quality, meal timing patterns, and daily light–dark cycles that entrain circadian rhythms [15,64,150,156,163].

Preclinical studies have shown that early meals [16,41], schedules of eating–fasting aligned with active–resting phases [10,11,14,160,164], and limiting daily food intake to a 6–10 h period during the active phase can restore the robust oscillation of CGs and improve metabolic outcomes in mouse models of circadian disruption and diet-induced obesity [10,11,12,13,14,41,65]. Notably, in these animal models, it was shown that TRE and its resetting effects on circadian CGs positively influence GM oscillation, composition, and function [14,15,65,150]. Furthermore, the TRE-induced upregulation of GM and CG oscillation was strongly associated with improved glucose tolerance, insulin sensitivity, and better glycemic control [11,12,13,14,41,63,65,150].

These improved cyclical changes in the GM resulting from the time-dependent eating patterns, i.e., TRE where food is available only during the nocturnal active phase, may contribute to the diversity of gut microflora and represent a mechanism by which the gut microbiome may reduce gut permeability and systemic inflammation that enhance metabolic responses to environmental challenges, especially to different meal timing patterns [14,63,64,150,156,163].

Interestingly, another study found that mice under late time-restricted feeding (late TRE) displayed asynchrony in their liver CGs and experienced disrupted metabolism. However, when the feeding window was shifted to the active phase (early TRF), circadian CGs were restored, leading to metabolic benefits [164]. Moreover, circadian CG rhythms were maintained even when early TRE was implemented in disruptive light conditions. This suggests that the timing of food (meal timing alignment) is the dominant driver influencing peripheral CGs’ rhythmicity. This also implies that early TRE can restore disrupted clock gene expression and prevent the harmful effects of misaligned meal timing and an HFD [10,11,14,41,60,61,65,164,165,166,167,168].

As previously discussed, clinical trials targeting early meals, including dietary interventions (DIs) that emphasize consuming most calories at breakfast, referred to as the breakfast diet [17,18,25,26,27,161,162], as well as a DI with three daily meals, starting with early breakfast at 8:00 [9,19,20,23,24,28], have been shown to have resetting effects and upregulate circadian CG expression [17,18,19,20] and are associated with more effective weight loss [18,25,26,27], improved glycemic control [9,18,21,22,23,24], better appetite regulation [18,21,26,162], and increased energy expenditure [28,37,38,39].

First, we will describe clinical studies comparing the effects of eating breakfast versus skipping it on CG expression and associated outcomes. Second, we will discuss the research evidence showing the beneficial effect of eating breakfast and consuming calories early in the day (breakfast diet) in the context of three daily meals versus six daily meals on circadian CG expression, weight loss, and glycemic regulation in obese and T2D individuals. Finally, we will describe preclinical and clinical studies that explain how, besides restoring the CG rhythms, the aligned eating–fasting schedule may positively affect gut microbiota oscillation, composition, and function.

### 5.1. Effect of Eating Versus Skipping Breakfast on Circadian CG mRNA Expression and Glucose Excursions in T2D

It was revealed that skipping the day’s first meal, equivalent to delaying breakfast until noon, disrupts and blunts the rhythmicity of CG expression in animal models [16,40,169], which was associated with increased body weight and decreased muscle weight [40,169]. Noteworthy is that the insulin response after CH intake is enhanced in the early hours of the day, highlighting the essential role of an early high-energy breakfast as a resetting signal for circadian CGs and regulation of glucose metabolism [29].

Clinical and epidemiological studies in humans have reported that omitting or delaying breakfast is associated with increased body weight, appetite, higher overall glycemia, and HbA1c [44,45,46,47,169,170].

To determine whether skipping breakfast versus breakfast consumption influences the glycemic response after an isocaloric lunch and dinner in humans, we conducted a single-day randomized crossover study in T2D participants [44]. We found that the omission of breakfast (NoB) and prolonged fast over 16 h until noon, as opposed to consuming a high-energy breakfast (YesB) just after the overnight fast, led to significantly higher glycemic responses and deficient and delayed insulin, C-peptide, and iGLP-1 responses after subsequent isocaloric lunch and dinner [44]. These results are consistent with previous studies showing that skipping breakfast leads to higher glycemic response and deficient insulin response after subsequent meals [17,86,169,171].

Next, we investigated whether the disruption of CG expression induced by breakfast omission was the underlying cause of the increased glycemic response and deficient insulin response after subsequent meals [17]. To this end, we conducted another single-day randomized crossover study on healthy and T2D participants to explore the impact of NoB versus YesB on the expression of metabolic CGs. The omission of breakfast (NoB) acutely disrupted CG expression by downregulating *AMPK*, *BMAL1*, *PER1*, and *RORα* mRNA expression after lunch [17]. This circadian CG disruption on the NoB day was associated with significantly higher glycemic response, deficient and delayed insulin, and intact GLP-1 post-lunch responses [17]. Importantly, the negative effect of omitting breakfast was wholly reversed on the day breakfast was consumed. Indeed, cutting the overnight fast at 8:00 by consuming a high-energy breakfast on YesB day led to a resetting effect of these key metabolic CGs, a significant reduction in postprandial glycemia, and enhanced and faster insulin and GLP-1 responses after lunch [17].

In particular, the upregulation of *AMPK* on YesB day significantly enhances GLUT-4 translocation, muscular glucose uptake, and postprandial insulin response, leading to reduced post-meal glycemic excursions [17,82,84,85]. *AMPK* is also positively linked to *SIRT1* and its beneficial effects on insulin sensitivity, β-cell proliferation, and viability [83].

These two studies indicated that skipping breakfast and prolonged fasting until noon can disrupt the expression of CGs and lead to a significantly higher glycemic response, deficient and delayed insulin, and intact GLP-1 after subsequent meals in healthy individuals and those with T2D [17,44]. Moreover, the beneficial effect of breakfast consumption was shown in studies on TRE, which reported that TRE shifted to the early hours of the day (early TRE) led to maximal improvement of glucose metabolism in healthy, obese, and T2D subjects [7,8,9,20,23,42].

Many studies in rodents [16,40,169] and humans [17,45,46,170] have shown that the omission of or delayed breakfast is directly associated with disrupted circadian CG expression [16,17] and increased body weight, appetite, overall glycemia, and HbA1c [44,45,46,47,169,170].

Some authors, however, reported that skipping dinner may have similar or even worse detrimental effects than omitting breakfast [48]. In a retrospective cohort study on 17,573 male and 8860 female university students during a three-year observation period, it was shown for both men and women that skipping dinner, not breakfast or lunch, was associated with overweight or obesity [48]. Another cross-sectional study involving 14,286 students found that skipping meals was associated with several confounding factors, including socioeconomic status, social inequalities such as maternal illiteracy, unemployment, and other lifestyle factors. Therefore, it is challenging to determine the causal relationship between skipping meals and detrimental health outcomes [49]. Furthermore, another study that included 14,400 children and adolescents aged 7–18 discovered that skipping meals, whether breakfast, lunch, or dinner, was linked to physical and mental health issues, such as headaches, stomachaches, difficulty falling asleep, and nervousness. Therefore, having regular meals at least three times a day is important in this population [50]. A study in low-income Latinos with poorly controlled T2D showed that skipping breakfast or dinner is linked to poor glycemic control. However, this association should be interpreted cautiously, as lower socioeconomic status significantly determines fasting plasma glucose and HbA1c levels [51].

### 5.2. Effect of 3M Diet and Shifting Most Calories to the Early Hours of the Day (Breakfast Diet) on Circadian CG Expression and Energy and Glucose Metabolism in T2D

Asynchrony of peripheral CG expression is an essential feature of T2D in rodents [29,95,157] and humans [17,18,100,157]. Circadian CG disruption has been associated with insulin resistance, reduced and delayed β-cell secretion, increased hepatic glucose output, and low muscular glucose uptake [17,18,29,86,157,159].

Traditionally, dietary interventions (DIs) for T2D involve consuming five or six small daily meals with calories and CH spread evenly. However, studies conducted on rodents have indicated that frequent meals, especially when consumed without time restrictions or proper alignment—such as during resting hours—can lead to weight gain, disturbance of glucose and lipid metabolism, and further asynchrony of circadian CGs, which are usually disrupted in T2D [10,11,12,13,14,15,63]. Moreover, enhanced insulin response after CH intake in the early hours of the active phase (breakfast) has been documented as an essential resetting signal for circadian CG expression [29].

Both in healthy and T2D individuals, frequent small meals evenly distributed throughout the day, along with skipping breakfast and overeating in the evening, have been associated with weight gain, poorly controlled glucose, and higher HbA1c compared to breakfast consumption in the context of two or three daily meals [45,46,162]. In addition to the beneficial effect of reducing meal frequency to only two or three daily meals, early meals and the assignment of most calories and CH shortly after waking (breakfast diet) have been associated with effective weight loss [26,27], improved insulin sensitivity, β-cell secretion, and glycemic control [7,8,9,21,23,24,25,26,162], and decreased hunger scores [21,26,172].

Therefore, we investigated whether a high-energy and CH breakfast in the context of three daily meals and reducing CH intake in the evening were more beneficial for aligning circadian CG rhythms and glucose metabolism in individuals with T2D compared to consuming the same amount of CH spread evenly across six meals throughout the day [18]. We studied the effects of 12 weeks on two different isocaloric DIs—HB-3M diet (*n* = 18) or 6M diet (*n* = 1735)—on T2D participants treated with insulin. Those on the HB-3M diet ate three meals: a large breakfast of 700 kcal, a medium-sized lunch of 600 kcal, and a small dinner of 200 kcal. They were instructed to have breakfast before 9:30, lunch between 12:00 and 15:00, and dinner between 18:00 and 20:00. The 6M diet participants ate six meals, including breakfast, lunch, and dinner, at the same times as those on the 3M diet. In addition to these main meals, they had three snacks scheduled at 11:00, 17:00, and 22:00, with daily calories and CH intake evenly distributed throughout the day.

In comparison to the 6M diet, the HB-3M diet led to a significant resetting effect in CG oscillatory expression, i.e., in *BMAL1*, *CRY1*, and *PER2*, which are linked to the regulation of muscular glucose uptake, hepatic glucose production, and insulin secretion. Likewise, the HB-3M diet significantly increased the mean daily *RORα* and *SIRT1* levels, which are involved in B-cell secretion, GLUT4 muscular glucose uptake, and lipolysis [18]. This enhanced CG synchronization with the HB-3M diet was associated with greater weight loss, reduction in HbA1c, fasting glucose, and significant decreased in glucose excursions and mean glucose assessed over 24 h by continuous glucose monitoring (CGM). It reduced the total daily insulin dose (TDID) in most HB-3M diet participants [18]. Notably, the overall glucose excursions with the HB-3M diet were significantly reduced during the nocturnal segment (00:00 to 06:00). This suggests a reduction in nocturnal hepatic glucose production and improved hepatic insulin sensitivity with the HB-3M diet [35].

We concluded that a three-meal DI, with most calories and CH assigned to the early hours of the day, may be a beneficial approach for optimizing circadian CG regulation of glucose homeostasis in obese and T2D individuals [18].

### 5.3. The Interaction Between Meal Timing Schedule, Circadian CG mRNA Expression, and the Gut Microbiome (GM) Is Essential in Regulating Energy and Glucose Metabolism

Molecular circadian CGs and GM relationships evolved as mechanisms that enhance metabolic responses to environmental challenges. Many studies have assessed the influence of meal timing and feeding–fasting schedule on the bidirectional interactions between circadian CGs, the GM, and metabolism and their effects on obesity and energy homeostasis [14,53,54,64,150]. Circadian CGs coordinate daily changes in the composition and function of the gut microbiota (GM). In turn, GM-derived bioactive metabolites provide feedback to circadian CGs and metabolism in animal models [53,62,63,64,65,102,164,165,166,167,168,173] and in human studies [54,60,61,62,64,150].

The feeding–fasting schedule critically influences circadian CG expression and metabolism in mice [11,14,16,167,168,173] and humans [17,18,19,157]. Furthermore, the meal schedule plays an interactive role between CGs and the GM, establishing an essential connection in regulating energy and glucose homeostasis [14,54,65,102,150,164,166]. As a result of the meal timing–CG–GM interaction, the microbiota exhibits diurnal oscillations influenced by feeding–fasting rhythms, leading to mealtime-specific changes in the GM throughout the day in mice [52,60,61,62,63,166] and humans [52,60,61,62]. Hence, the GM abundance and alpha diversity are generally higher during the active or feeding phase and lower during the rest or fasting phase [63,64,65,102,103,104].

It was documented that misaligned meal timing may cause simultaneous asynchrony of circadian CGs and GM oscillation associated with insulin resistance and abnormal glucose and lipid metabolism [11,14,53,63,66,150].

Interestingly, studies linking the GM to obesity, insulin resistance, metabolic syndrome, and diabetes have found that fecal transplantation demonstrates its pivotal role in regulating insulin resistance and inflammation [63,135,174]. Moreover, studies that examined the effects of experimental jet lag in mice and humans [63] showed that misaligned meal timing induced by jet lag led to disrupted CG expression, which in turn caused abnormal fluctuation and dysbiosis of the GM, along with an imbalance linked to glucose intolerance and obesity [63]. Furthermore, the dysbiosis induced by the experimental jet lag was transferable through fecal transplantation, resulting in similar disruptions to glucose intolerance and obesity [63,174].

Many other studies have shown that circadian CG machinery is essential for maintaining daily fluctuations in the GM, which is critical for regulating energy and glucose metabolism and immune responses, in animal models [63,64,134,136,137,138,139,167] and also in some human studies [140,142,143,144].

Research evidence has demonstrated that disruptions to PER1 and PER2, as observed in PER1/2−/− mice lacking a functional clock, resulted in a significant or complete loss of the rhythmic fluctuations of the GM compared to wild-type mice [63,64,168,173]. It was also noted that PER1/2-deficient mice had GM dysbiosis, lower alpha diversity, and a different intestinal community composition compared to wild-type mice [63,64,168,173]. As mentioned previously, the eating–fasting schedule and TRE, along with influencing circadian CG expression, play a significant role in establishing the rhythms of the GM and the diurnal fluctuation in operational taxonomic units (OTUs) in animal models. This highlights the importance of meal timing and the regulation of GM oscillation [63,64].

Interestingly, the phases of GM oscillations are closely aligned with the feeding–fasting schedule in both wild-type and PER1/2−/− mice [63,64]. This indicates that the feeding–fasting schedule dictates daily GM oscillations even if a functional clock is lacking. Therefore, GM oscillation is a flexible process that can be lost (as observed in PER1/2−/− mice) or restored in response to changes in the feeding–fasting schedule [63,64,168,173].

The impact of a high-fat diet (HFD) on circadian CGs and the GM has also been extensively researched [11,12,14,65,165,166]. Mice with unrestricted or ad libitum access to an HFD throughout the day (AL-HFD), along with disrupted circadian CG expression, also exhibited reduced activity of GM, changes in microbial diversity, and loss of synchronization between feeding rhythms and GM oscillation [11,12,14,65,166], and their GM contains half as many cycling OTUs [14]. AL-HFD mice also developed diet-induced obesity (DIO) and metabolic disorders, including hyperinsulinemia, hypercholesterolemia, high triglycerides, hepatic steatosis, and impaired glucose regulation [11,14,65,165,166].

In contrast, HFD mice subjected to time-restricted feeding during their early active phase (early TRE-HFD) showed restored molecular rhythms of *CLOCK* and *CRY1*, phase-advanced expression of *PER1*, *PER2*, *CRY2*, *BMAL1*, *RORα*, and *REV-ERBα*, and recovery of GM oscillation [11,12,14,65,165,166]. Furthermore, early TRE-HFD protects against AL-HFD-induced obesity, insulin resistance, hyperlipidemia, hepatic steatosis, and other metabolic disorders associated with HFD [11,12,13,14,15,65,166,167]. Early active-phase TRE-HFD also increases *AMPK* and *SIRT1* activity in the liver, which mediates the benefits of meal-timed restriction [11,12,14,84,85].

Another study found that mice under late time-restricted eating (late TRE) displayed asynchrony in their liver CGs and experienced disrupted metabolism. However, when the feeding window was shifted to the active phase (early TRE), the circadian CGs were restored, leading to metabolic benefits [164]. Moreover, circadian CG rhythms were maintained even when early TRE was implemented in disruptive light conditions. This suggests that the timing of food (meal timing alignment) is the dominant driver influencing peripheral CG rhythmicity. It also implies that early TRE can restore disrupted clock gene expression and prevent the harmful effects of misaligned meal timing and HFD [10,11,14,41,60,65,164,165,166,167,168].

Furthermore, studies have demonstrated that wild-type mice, whether fed during the dark or light period, exhibited phase-reversed GM oscillation and microbial mucosal attachment rhythms [11,12,13,14,15,165,166,168]. These findings identify feeding time as a major driver of GM oscillation.

To further assess the effects of TRE in humans, several clinical studies were conducted during Ramadan. One of these studies included healthy nondiabetic Ramadan practitioners (18 males, five females; mean age 23.1 years) who were evaluated before and two weeks into Ramadan. Blood was collected for measurements of endocrine and metabolic parameters at 9 a.m. (±1 h) and again twelve hours later. Compared to pre-Ramadan values, glucose concentration was kept within the normal range, circadian cortisol rhythm was abolished, and cortisol levels continuously increased, with evening hypercortisolism. Ramadan practitioners have altered adipokine patterns typical of insulin resistance, i.e., leptin levels were significantly increased. In contrast, adiponectin levels were significantly lower. These changes were associated with increased insulin resistance in the morning and evening. hsCRP levels decreased and there was a loss of circadian rhythmicity of hsCRP, probably due to the loss of circadian cortisol rhythm. All are likely to increase cardiometabolic risk [175].

In a subsequent study conducted during Ramadan by the same group [176] on healthy young volunteers (5 females, 18 males; mean age 23.2 years), it was found that concomitant with an abolished circadian rhythm of cortisol and continuously elevated cortisol levels, increased leptin and low adiponectin levels favoring insulin resistance, there was disrupted mRNA expression and levels of CLOCK and glucocorticoid-controlled genes, such as DUSP1 and IL-1α, as assessed in circulating leukocytes. There was a significant decrease in CLOCK expression in the morning and significantly higher DUSP1 mRNA expression and levels in the morning than in the evening. However, diurnal rhythm was maintained. Morning IL-1α mRNA expression remained significantly higher than in the evening during Ramadan, but was markedly decreased compared to pre-Ramadan levels. Likewise, the diurnal rhythm of hs-CRP was lost, but the mean levels of hs-CRP were decreased. The improvements in some cardiometabolic risk factors, such as decreased circulating hs-CRP and IL-1α mRNA expression, suggest that intermittent fasting might have a beneficial component [176].

An additional study aimed to research the effect of TRE or intermittent fasting during Ramadan on gut microbiota [177]. The study was conducted on 12 healthy adult individuals who practiced fasting 17 h per day for 29 consecutive days during Ramadan. The correlations between the dietary intake measurements of the participants and the respective relative abundance of GM bacterial genera were investigated. The results showed that *Firmicute*s was higher in abundance before Ramadan and significantly lower at the end of Ramadan. *Proteobacteria* was significantly higher in abundance at the end of Ramadan. The fasting month resulted in a significant decrease in levels of seven genera, including *Faecalicatena*, *Fu*s*icatenibacter*, and *Lachnoclo*s*tridium*. Conversely, the abundance of two bacterial genera, *E*s*cherichia and Shigella*, was enhanced at the end of Ramadan. The results suggest that even when the fasting period during Ramadan was consistent, the types of food consumed by individuals can also influence GM composition and variability [177].

In a recent multicenter study, adult males who underwent time-restricted eating (TRE) showed increased GM diversity and upregulation of circadian CGs, namely, *BMAL1* and *CLOCK* mRNA expression, compared to a control group without time restrictions on eating. Additionally, the TRE group experienced significant reductions in total cholesterol and triglyceride levels and increased high-density-lipoprotein cholesterol [72,73].

Notably, the positive changes in GM observed in preclinical studies with time-restricted eating (TRE) and/or by adjusting calorie intake to the active phase were linked to restored molecular phases of *CLOCK* and *CRY1* and advanced phases of *PER1*, *PER2*, *CRY2*, *BMAL1*, *RORα*, and *REV-ERBα* [10,11,12,13,14,153,167,168,173].

This suggests that adjusting feeding and fasting schedules may significantly influence circadian CGs and GM, improving energy, lipid, and glucose metabolism outcomes.

## 6. Conclusions and Remarks

The circadian clock synchronizes the diurnal oscillations of the metabolic processes involved in obesity and T2D, including β-cell secretion and muscular glucose uptake. CGs in the central and peripheral clocks are entrained with the light–dark cycle and food cues, namely, the feeding–fasting schedule.

Meal timing, especially breakfast consumption, has a powerful resetting effect on circadian CGs and metabolic outputs. Aligned meal timing, such as high-energy breakfast and reduced intake in the evening, can upregulate the CGs involved in hepatic glucose production, insulin secretion, and muscle glucose uptake. This approach is associated with significant improvements in weight loss and glycemic outcomes, providing a promising strategy for managing obesity and T2D.

The feeding and fasting schedule also powerfully synchronizes GM oscillation. The GM may modulate the dynamic interaction between the feeding and fasting schedule and circadian CGs.

In coordination with the feeding–fasting schedule, the GM consumes the nutrients from the ingested food, and the resulting metabolites and bioactive molecules are released into the bloodstream. These GM-derived metabolites, such as short-chain fatty acids (SCFAs), products of bile acid (BAs) metabolism, lipopolysaccharides (LPSs), interleukins, and proinflammatory cytokines, play a crucial role in insulin resistance, muscle glucose uptake, and lipid metabolism. Furthermore, GM-derived bioactive molecules also signal circadian CGs, controlling energy and glucose metabolism.

Misaligned meal timing, such as skipping breakfast or snacking all day, along with asynchrony in metabolic CG expression, also lead to parallel disruption of GM (dysbiosis) and are associated with hyperglycemia, insulin resistance, obesity, and diabetes. This provides evidence of the GM-dependent mechanism for common metabolic disturbances in humans with aberrant feeding rhythms, as seen in shift workers or those who usually omit breakfast. In contrast, aligned meal timing, i.e., shifting most calories to the early hours of the day, restores the oscillation of GM and CG expression and improves weight loss, energy, and glucose metabolism. More studies are needed to explore the underlying mechanism of these deleterious effects of misaligned meal timing on GM, CG expression, and metabolism. It appears that consuming more calories earlier in the day (i.e., early breakfast) may confer a metabolic advantage, especially in the context of weight loss and overall glycemia, by coordinating the reciprocal cross talk between gut microbiota, circadian clock genes, and food intake to coincide with optimal circadian timing.

## Figures and Tables

**Figure 1 ijms-25-12355-f001:**
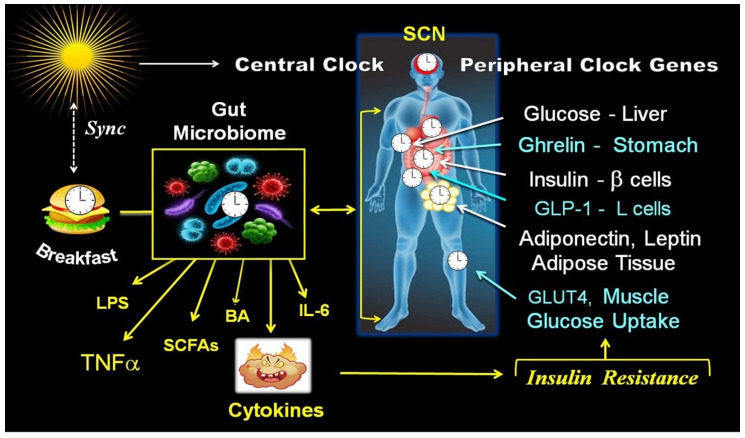
Interactions among meal timing, circadian clock genes, and gut microbiome. The timing of food intake significantly impacts how the GM regulates the expression of circadian CGs and energy and glucose metabolism. As we can see in the illustration, the meal timing coordinates the rhythm of the GM and the release of GM-derived metabolites such as short-chain fatty acids (SCFAs), lipopolysaccharides (LPSs), bile acids (BAs), interleukin 6 (IL-6), tumor necrosis factor alpha (TNF-α), and other cytokines. These metabolites in turn can influence circadian CG expression and the body’s hormonal and metabolic response to meal timing.

**Figure 2 ijms-25-12355-f002:**
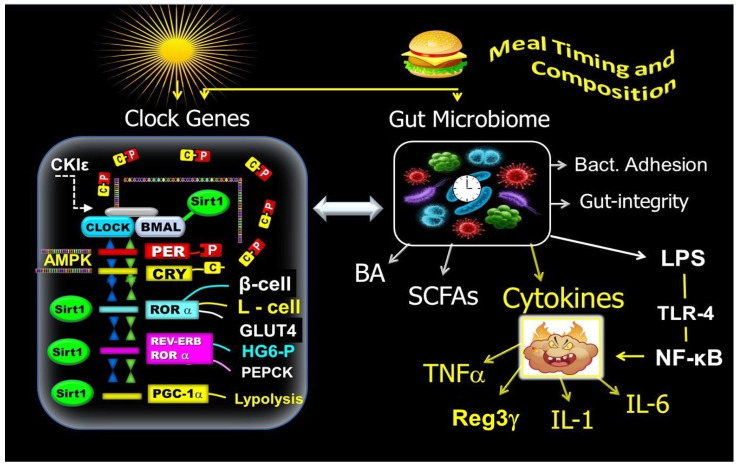
Mechanism of meal timing interactions with molecular CGs and the gut microbiome. The illustration shows on the left how the meal timing and light at dawn activate the CLOCK–BMAL1 complex associated with *SIRT1* and the transcription of *PER*s and *CRY*s to form PER (P) and CRY (C) proteins. PER and CRY proteins form dimers (CP) in the cytoplasm. Subsequently, the CP dimers translocate back to the nucleus to repress CLOCK–BMAL1. The blockage of CLOCK–BMAL1 is reversed by casein kinase I epsilon (CKIε), restarting a new cycle. In addition, AMPK positively interacts with *SIRT1*. CLOCK–BMAL1 complex-driven transcription of *PER*s, *CRY*s, *REV-ERBα*, and *RORα* genes, and *PGC-1α*, promote the expression of tissue-specific clock-controlled genes. This upregulates β-cell insulin secretion, L-cell postprandial incretin GLP-1 response, increase in muscle GLUT4 activity, and glucose uptake. The clock gene *REVERBα*-, *RORα*-, and *SIRT1*-driven nocturnal hepatic glucose production in the liver promotes glycogenolysis enzymes HG6-P and PEPCK of the gluconeogenesis pathway. In addition. *BMAL1* transcription of *PPARα* and *PGC-1α* in adipose tissue plays a role in nocturnal lipolysis. On the right is a representation of how the GM, through the secretion of SCFAs, influences BA metabolism and gut permeability, and its influence on the release of LPSs and proinflammatory cytokines, including, TNF-α, Reg3γ, IL-1, and IL-6, may exert modulatory effect on CG expression and subsequently influence energy and glucose metabolism. Abbreviations: Glucose 6-phosphatase: HG6-P; phosphoenolpyruvate carboxykinase: PEPCK; short-chain fatty acids: SCFAs; bile acids: BA; lipopolysaccharides: LPSs; interleukin 1: IL-1; interleukin 6: IL-6; tumor necrosis factor α: TNFα; regenerating islet-derived protein III gamma: Reg3γ; Toll-like receptor: TLR.

**Figure 3 ijms-25-12355-f003:**
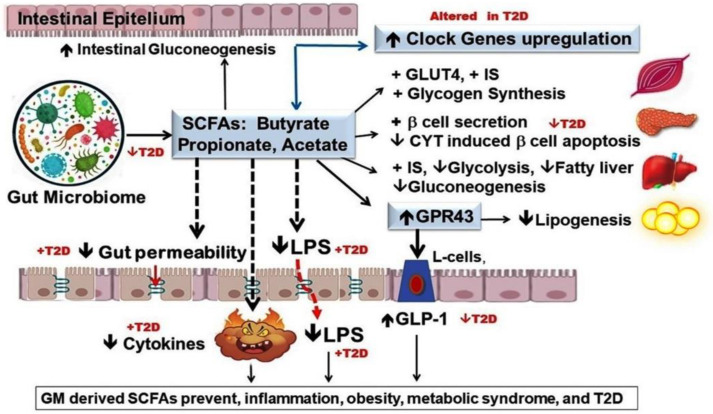
Role of GM-derived SCFAs in obesity, metabolic syndrome, inflammation, and T2D. The diagram illustrates how butyrate, propionate, and acetate, the three main SCFAs, help reduce gut permeability and have an anti-inflammatory effect by decreasing the release of cytokines and lipopolysaccharides (LPSs). SCFAs also influence glucose metabolism by binding to specific cell-surface G protein-coupled GPR43/41 receptors in various tissues. They also increase muscle GLUT4, enhance muscular glucose uptake, and promote glycogen synthesis. Moreover, SCFAs improve insulin sensitivity and β-cell secretion while decreasing glycolysis and gluconeogenesis in the liver. Activation of GPR43/41 by SCFAs reduces lipogenesis in adipose tissue and stimulates the secretion of GLP-1. The effects of SCFAs disrupted in T2D are highlighted in red.

**Figure 4 ijms-25-12355-f004:**
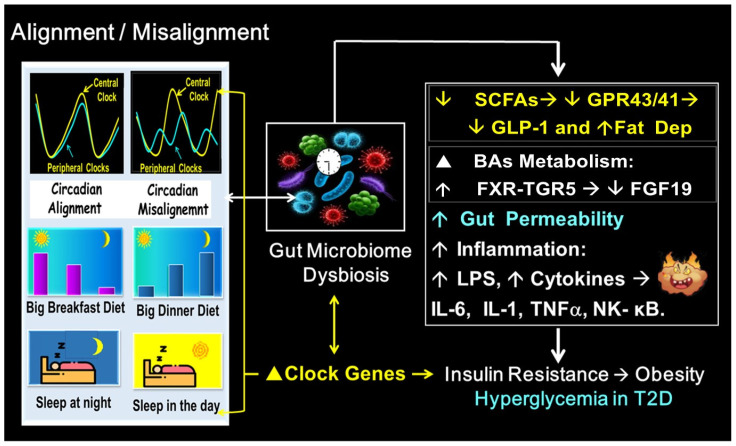
Effects of misalignment on circadian clock genes, gut microbiome, and energy and glucose metabolism. In this illustration, we observe on the left how eating and sleeping hours are not aligned with the circadian clock, i.e., small breakfast, big dinner, and sleeping during daylight lead to disrupted CG expression and GM dysbiosis. On the right, we observe several cellular mechanisms by which GM dysbiosis may cause insulin resistance, obesity, and hyperglycemia in T2D. First, GM dysbiosis leads to reduced release of SCFAs and binding to the cell-surface GPR43/41 receptor. This results in less secretion of GLP-1 from enteroendocrine L-cells and to increased insulin-mediated fat accumulation. GM dysbiosis causes a reduction in secondary BAs, deficient binding of BAs with FXR and TGR5 interaction, and a reduction in FGF19 secretion, diminished energy expenditure, and abnormal glucose homeostasis. Lower binding of secondary BAs to TGR5 to form the BA–TGR-5 complex may also lead to a reduction in GLP-1 secretion from intestinal L-cells and a reduction in glucose-stimulated insulin release from pancreatic β-cells. More importantly, GM dysbiosis enhances the secretion of cytokines, i.e., Il-6, IL-1, hs-CRP, LPSs, TNFα, and nuclear factor kappa-light-chain-enhancer of activated B cells (NF-κB), all of which activate several pathways that worsen insulin resistance, obesity, and hyperglycemia.

## Data Availability

Not applicable.

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
