# Peer review of "Interaction Between Early Meals (Big-Breakfast Diet), Clock Gene mRNA Expression, and Gut Microbiome to Regulate Weight Loss and Glucose Metabolism in Obesity and Type 2 Diabetes"

_ijms, 2024, doi:10.3390/ijms252212355_

Round 1

Reviewer 1 Report

Comments and Suggestions for Authors

The authors provide a narrative review on the interplay of the gut microbiome, circadian rhythm and early meal timing in the context of obesity and T2D as well as their dietary treatment.

The overall rationale of the paper is clear.

Some aspects warrant further clarification:

The authors claim, that breakfast skipping as a unique detrimental effect on human metabolism, based on observational studies. First, these observational studies cannot provide evidence for causality and might either suffer from reverse causality or (residual) confounders such as socioeconomic status and other lifestyle factors. Also, similar or even stronger detrimental associations are reported for dinner skipping. (https://pubmed.ncbi.nlm.nih.gov/33477859/; https://pubmed.ncbi.nlm.nih.gov/34877008/, https://pubmed.ncbi.nlm.nih.gov/30155854/, https://pubmed.ncbi.nlm.nih.gov/21181446/). Other studies show beneficial associations for both early and late TRF: https://pubmed.ncbi.nlm.nih.gov/36971368/.

The very few human RCTs comparing early and late TRE should be demonstrated in detail.

Furthermore, your paper needs to more clearly distinguish between data from rodent and human trials.

Sections 3.3.1-3.3.5 are full of correct content, but should be shortened as they do not contain material fully relevant for the main topic of the paper. Section 3.3.6. should be extended instead. Also, 3.3.6. mainly reports effects on or parallel to the gut microbiome, not those directly elicited BY the gut microbiome.

The "mays, coulds and mights" of your conclusion are very necessary, as the current state of the literature is way too sparse, inconsistent and indirect to formulate solid recommendations on "when to eat what".

Author Response

Response  to  Reviewer 1  IJMS (ISSN 1422-0067) Manuscript Ijms-3214678

Prof. Jakubowicz Daniela

From  the Reviewer 1

Comments and Suggestions for Authors

 Reviewer: The authors provide a narrative review on the interplay of the gut microbiome, circadian rhythm and early meal timing in the context of obesity and T2D as well as their dietary treatment. The overall rationale of the paper is clear.

Response: 

From  the Reviewer 1

Comments and Suggestions for Authors

Reviewer: The authors provide a narrative review on the interplay of the gut microbiome, circadian rhythm and early meal timing in the context of obesity and T2D as well as their dietary treatment. The overall rationale of the paper is clear.

Response: We highly appreciate the reviewer for the insightful comments, suggestions, and additional references, which will improve the clarity and accuracy of the scientific statements in our manuscript. To display all the modifications in the manuscript, we provide two attachments:  the clean final version and the draft version showing all the corrections made throughout the manuscript. The added parts are written in red, the deleted parts are shaded in grey,  and the distinction between  animal  and  clinical  data  is written  in violet

Reviewer: Some aspects warrant further clarification:

The authors claim, that breakfast skipping as a unique detrimental effect on human metabolism, based on observational studies

Response:  Thanks to the reviewer for this comment. Indeed, we need to clarify better the main endpoint of this manuscript.  The primary endpoint of this review is to show the effect of meal timing, especially the effect of high-energy breakfast, on Clock Genes (CGs) expression and collateral outcomes.  Therefore, in the introduction (paragraph # 3), we emphasized that breakfast exerts a powerful resetting effect (known as the "Zeitgeber effect") on the CGs network. In the same paragraph, we deleted the sentences on TRE and early time-restricted eating), as the clinical studies showing any influence of TRE  on circadian CG expression are very scarce. However, we made additional comments on the effect of TRE  later in the introduction (paragraph #7).

Reviewer: First, these observational studies cannot provide evidence for causality and might either suffer from reverse causality or (residual) confounders such as socioeconomic status and other lifestyle factors. Also, similar or even stronger detrimental associations are reported for dinner skipping. (https://pubmed.ncbi.nlm.nih.gov/33477859/; https://pubmed.ncbi.nlm.nih.gov/34877008/, https://pubmed.ncbi.nlm.nih.gov/30155854/, https://pubmed.ncbi.nlm.nih.gov/21181446/).

Response:  Thanks, the reviewer, for pointing out that are some discrepancies in  several  studies that  do not support the pivotal role of skipping breakfast in detrimental metabolic outcomes. Thanks also for providing references for the discussion of an inconsistences in studies analyzing the effect of skipping  meals. 

To address this discussion, first we explained in the introduction (paragraph # 5) the beneficial effects of eating versus skipping breakfast on circadian CGs expression and metabolic outcomes.

After that, also  in the introduction in paragraph #6 (written in green), we cited the studies recommended by the reviewer which  will  be analyzed to clarify some discrepancies related to the effect of skipping meals. These trials include the study by Yamamoto R et al., 2021 [49], https://pubmed.ncbi.nlm.nih.gov/33477859 , which reported similar or even worse detrimental effects caused by skipping dinner versus breakfast [49]. In addition, other studies show that the adverse effects of the omission of meals need to be analyzed with caution as there might be causality confounders, such as socioeconomic status, age, presence of T2D, and other lifestyle factors that may add collateral influence to the health outcomes. It  includes a study by  Qorbani M et al., 2021 [50], https://pubmed.ncbi.nlm.nih.gov/34877008/, the trial of Azemati B et al., 2020 [51], https://pubmed.ncbi.nlm.nih.gov/30155854 , and another study of Kollannoor-Samuel G et al., 2011 [52] https://pubmed.ncbi.nlm.nih.gov/21181446  which correlates the effects of skipping meals with glycemic levels and HbA1c.

         Please refer to section 5.2, starting in paragraph #6, where we discussed (in green) all the above cited studies including: Yamamoto et al. [49] showed that skipping dinner, not breakfast or lunch, was the most associated with overweight or obesity. In the study of Qorbani et al. [50 ], it was  documented  that skipping meals is  associated  with several confounders, such as socioeconomic status and other lifestyle factors, which make it difficult to interpret  the causality  between skipping meals and detrimental outcomes. In an additional study by Azemati et al. [51]  it was found that skipping meals, regardless of whether it is breakfast, lunch, or dinner, was associated with somatic and psychological health complaints. Finally, the study of  Kollannoor-Samuel et al. [52] conducted on low-income Latinos with poorly controlled T2D showed that skipping breakfast or dinner is associated with poor glycemic control. However, the  lower socioeconomic status plays a more significant role in determining glycemic control

Reviewer: Other studies show beneficial associations for both early and late TRF: The very few human RCTs comparing early and late TRE should be demonstrated in detail https://pubmed.ncbi.nlm.nih.gov/36971368/. Zeng at al. 2023;

Response:  there are only few clinical reports showing an effect of TRE on CGs expression and associated  metabolism: In the introduction paragraph #8, we described a study on TRE conducted in humans by Jamshed at al. [19]. This 4-day randomized crossover study, in healthy participants, explored the effects of early time-restricted eating (early TRE) with an eating window between 8 am and 2 pm versus an eating window from 8 am to 8 pm (control schedule). The early TRE led to upregulation of the markers of the circadian CGs and improved 24-hour glucose levels using continuous glucose monitoring (CGM) [19]]

Also in  the introduction paragraph #9 we quoted in green: Several other studies, however, reported a similar metabolic advantage of both early and late-time-restricted eating. [9, 23, 41, 43]  Indeed, in a cross-sectional study by  Zeng et al., 2023 [43], with  3813 participants. Both the early TRE (05.00-15.00) and late TRE (11.00-21.00) showed similar diminished risk of developing non-alcoholic fatty liver disease (NAFLD) [43].; https://pubmed.ncbi.nlm.nih.gov/36971368/

Reviewer: Furthermore, your paper needs to more clearly distinguish between data from rodent and human trials.

Response: Thanks for this observation. We differentiated the data from rodent and human trials in several sections across the manuscript. Please observe that the distinction between rodent and clinical studies results is written in violet.

It  can  be  observed namely:  in the introduction, paragraphs #3, #5, #6, #9, and #11. In the section 2.2, paragraph #3. In the section 3.1. paragraph # 4.  In the section 3.2. paragraphs #2, and #3. In the  section 5.2 paragraph #4. In the section 3.3.4 paragraph #2.  In the  section 5.2, paragraphs #5,and # 6 and in the  section 5.3 paragraphs #2, #3, #8, and #9

Reviewer: Sections 3.3.1-3.3.5 are full of correct content, but should be shortened as they do not contain material fully relevant for the main topic of the paper. Section 3.3.6. should be extended instead. Also, 3.3.6. mainly reports effects on or parallel to the gut microbiome, not those directly elicited BY the gut microbiome.

Response: Thanks, the  reviewer for these observations and suggestions. According  the sections 3.3.1 to  3.3.5  were  shorted. The  deleted  paragraphs  are shaded in  grey and   the addition are written in  red. The section 3.3.6. was extended with  additional   information  related to the  effect of  GM  on the  circadian CGs.

Reviewer: The "mays, coulds and mights" of your conclusion are very necessary, as the current state of the literature is way too sparse, inconsistent and indirect to formulate solid recommendations on "when to eat what".

Response: Thanks  for  this  suggestion.  We added   "mays, coulds and mights"  in  many of the  statements and  conclusion

Reviewer 2 Report

Comments and Suggestions for Authors

When I got the manuscript for review, I recalled the former Rector of my University confessing that he started to lose weight after  he had started to take breakfast. Many studies have provided scientific background to such phenomenon.

The manuscript presents a well and competently written review the big breakfast diet), expression of clock genes  and gut microbiome, weight loss and glucose metabolism in obese and Type 2 diabetic persons. The review emphasizes the different zeitgebers for the central clock and peripheral clocks and covers key aspects of these interrelated phenomena, including gut microbiome as an important player.

Remarks:

Are there any relevant epidemiologic data concerning comparison of populations with habits of heavy breakfast like in England and with light breakfast and main dish in the evening like in Spain or Italy?

I wonder whether „mRNA” in the title is necessary? Gene expression is usually studied by masurement of the mRNA production.

Line 173: is „namely” necessary?

Lines 272/273: „by increasing the expression of GLUT4 through AMP-activated protein kinase (AMPK) activity”, do the authors mean increased biosynthesis of the protein or enhanced translocation of the transporter to the plasma membrane (increased membrane expression)?

Lines 277 and 537:    beta cells?

Line 474: „byincreased”, please separate

Line 690: the abbreviations (SCFAs etc.) do not seem necessary here; they have been introduced before and are not used later on.

Author Response

When I got the manuscript for review, I recalled the former Rector of my University confessing that he started to lose weight after  he had started to take breakfast. Many studies have provided scientific background to such phenomenon.

The manuscript presents a well and competently written review the big breakfast diet), expression of clock genes  and gut microbiome, weight loss and glucose metabolism in obese and Type 2 diabetic persons. The review emphasizes the different zeitgebers for the central clock and peripheral clocks and covers key aspects of these interrelated phenomena, including gut microbiome as an important player.

Response: We are very grateful to the reviewer for the encouraging words and the thoughtful critique, which are very helpful in improving the accuracy of this manuscript.

Remarks:

Reviewer: Are there any relevant epidemiologic data concerning comparison of populations with habits of heavy breakfast like in England and with light breakfast and main dish in the evening like in Spain or Italy?

Response: Thank you. I believe the key epidemiological data comes from the studies by Mekary et al., 2012 [46] and Reutrakul et al., 2014 [47]. These studies focus on the impact of skipping breakfast, eating frequency, and snacking on the body weight and glycemic control of obese individuals and those with type 2 diabetes.

Reviewer: I wonder whether „mRNA” in the title is necessary? Gene expression is usually studied by measurement of the mRNA production.

Response: In our studies on the effect of breakfast on clock genes by Jakubowicz D. et al., 2017 [17] and Jakubowicz D. et al., 2019 [18], we measured Clock Gene mRNA expression using RNA reverse transcription and Quantitative Real-Time PCR in white cells. Given that clock gene mRNA expression is the primary endpoint of the review, I guess it is important to write it in the Title.

Reviewer: Line 173: is „namely” necessary?

Response: In the sentence “the gluconeogenesis pathway namely during the second part of the nocturnal phase [56, 57]. The  word ” Namely”  was  deleted, and it appears  shaded  in  gray in the  Draft of Reviewer 2 corrections

Reviewer: Lines 272/273: „by increasing the expression of GLUT4 through AMP-activated protein kinase (AMPK) activity”, do the authors mean increased biosynthesis of the protein or enhanced translocation of the transporter to the plasma membrane (increased membrane expression)? 

Response: The  reviewer  is right,  we  modified  the  sentences  as  follows: SCFAs have a significant impact on blood glucose metabolism by increasing the expression and translocation of GLUT4 to the cell membrane through AMP-activated protein kinase (AMPK) activity, thereby enhancing muscular glucose uptake. [109-112].  It appears  corrected  in  the Draft  of  Reviewer 2 corrections

Reviewer: Lines 277 and 537:    beta cells?

Response:  corrected to  β-cell,  in  the draft

Reviewer:Line 474: „byincreased”, please separate

Response:  Thank  you,  the  word  was separated

Reviewer: Line 690: the abbreviations (SCFAs etc.) do not seem necessary here; they have been introduced before and are not used later on.

Response: Yes,  thank you for the suggestion. These abbreviations were deleted

Round 2

Reviewer 1 Report

Comments and Suggestions for Authors

The authors have revised their manuscript in accordance to the reviewer's points of criticism. The paper has been considerably improved.

However, there is still some improvement necessary.

Any chapter should follow the same hierarchy of studies: observational studies (humans), interventional studies (animals), interventional studies (humans) OR interventional studies (animals), observational studies (humans), interventional studies (humans).

The hypothesis on TRE effects can only be benefit (observational studies on Ramadan or animal studies) or harm (observational studies on Ramadan or meal skipping). If you assume benefit, start with the according studies, then show contradictions from similar studies and end up with high-quality (R)CTs in humans. If you assume harm, follow the according path. You already clarified in your manuscript, which studies were done in humans or animals; now, it's necessary to bring a more clearer order to the chapters. Otherwise, there is no transparent perspective on existing contradictions and their hierarchy of evidence.

Comments on the Quality of English Language

minor

Author Response

Responses to additional reviewer's suggestions to improve in the manuscript ijms-3214678  “Interaction between Early Meal Timing (Big Breakfast Diet), Clock Gene mRNA expression, and Gut Microbiome to Regulate Weight Loss and Glucose Metabolism in Obese and Type 2 Diabetes”.

Responses

            We are very grateful to the Reviewer for the detailed critiques and thoughtful suggestions aimed at improving the manuscript.

  • In reference to  the  first  point:

Reviewer: The paper has been considerably improved. However, some improvement is still necessary. Any chapter should follow the same hierarchy of studies: observational studies (humans), interventional studies (animals), interventional studies (humans) or interventional studies (animals), observational studies (humans), and interventional studies (humans).

Response: We are very grateful for this suggestion to follow a hierarchy of studies in all chapters. Yes, we addressed this point in each chapter of the draft. To show that we followed the suggested hierarchy, we highlighted in red what kind of studies we are referring to in each chapter section: usually → interventional studies (animals), observational studies (humans), and interventional studies (humans).

  • In reference to  the  second point:

Reviewer: The hypothesis on TRE effects can only be benefit (observational studies on Ramadan or animal studies) or harm (observational studies on Ramadan or meal skipping). If you assume benefit, start with the according studies, then show contradictions from similar studies and end up with high-quality (R)CTs in humans. If you assume harm, follow the according path. You already clarified in your manuscript, which studies were done in humans or animals; now, it's necessary to bring a more clearer order to the chapters. Otherwise, there is no transparent perspective on existing contradictions and their hierarchy of evidence.

Response:  Thanks to the reviewer for these comments. The studies on the effect of TRE on circadian CGs and GM were primarily conducted in rodents and have been addressed in section  5.3.  We also described the studies on TRE in humans and its effects on glycemic control, insulin resistance, lipids, and appetite. However,  most of these studies did not explore a concomitant effect of circadian CG expression and GM. We also added in section 5.3 three studies conducted during Ramadan: one focusing on metabolic outcomes,  another on the influence of Ramadan on circadian CGs, and a third on the effect of GM.  Similar to other TRE trials in humans, the simultaneous effects on CGs and  GM during Ramadan were not explored.

Other Specifications  across the manuscript:

--Specified subtitles in red in each  section to explain the order of  studies  we are describing, i.e.,  interventional studies (animals), observational studies (humans), interventional studies (humans), and clinical controversies in all of the sections along the manuscript.   In the clean version, these red subtitles were eliminated.

- Change: Section 5.2, “Effect of Eating versus Skipping Breakfast on Circadian CG mRNA Expression and Glucose Excursions in T2D, " was changed to Section 5.1. It is  also  specified in the text  of the corrected  draft.

-Change: Section 5.1, “Effect of 3Mdiet and Shifting Most Calories to the Early Hours of the Day (Breakfast Diet) on circadian CG expression and Energy and Glucose Metabolism in T2D,”   was changed to section 5.2. It is  specified in  the text  of the  corrected draft

-Paragraphs shaded in grey are eliminated paragraphs,  and

-Paragraphs in violet are additions.

Again, our  appreciation  to  the  reviewer  for the thoughtful  suggestions

Prof  Daniela  Jakubowicz  and  all   coauthors

Round 3

Reviewer 1 Report

Comments and Suggestions for Authors

Paper is ready for acceptance